# siRNA Treatment Enhances Collagen Fiber Formation in Tissue-Engineered Meniscus via Transient Inhibition of Aggrecan Production

**DOI:** 10.3390/bioengineering11121308

**Published:** 2024-12-23

**Authors:** Serafina G. Lopez, Lara A. Estroff, Lawrence J. Bonassar

**Affiliations:** 1Meinig of Biomedical Engineering, Cornell University, Ithaca, NY 14853, USA; sl3222@cornell.edu; 2Department of Materials Science and Engineering, Cornell University, Ithaca, NY 14853, USA; lae37@cornell.edu; 3Kavli Institute for Nanoscale Science at Cornell, Cornell University, Ithaca, NY 14853, USA; 4Sibley School of Mechanical and Aerospace Engineering, Cornell University, Ithaca, NY 14853, USA

**Keywords:** elastography, fibrocartilage, proteoglycans, RNAi, mechanics

## Abstract

The complex collagen network of the native meniscus and the gradient of the density and alignment of this network through the meniscal enthesis is essential for the proper mechanical function of these tissues. This architecture is difficult to recapitulate in tissue-engineered replacement strategies. Prenatally, the organization of the collagen fiber network is established and aggrecan content is minimal. In vitro, fibrochondrocytes (FCCs) produce proteoglycans and associated glycosaminoglycan (GAG) chains early in culture, which can inhibit collagen fiber formation during the maturation of tissue-engineered menisci. Thus, it would be beneficial to both specifically and temporarily block deposition of proteoglycans early in culture. In this study, we transiently inhibited aggrecan production by meniscal fibrochondrocytes using siRNA in collagen gel-based tissue-engineered constructs. We evaluated the effect of siRNA treatment on the formation of collagen fibrils and bulk and microscale tensile properties. Specific inhibition of aggrecan production by fibrochondrocytes via siRNA was successful both in 2D monolayer cell culture and 3D tissue culture. This inhibition during early maturation of these in vitro constructs increased collagen fibril diameter by more than 2-fold. This increase in fibril diameter allowed these tissues to distribute strains more effectively at the local level, particularly at the interface of the bone and soft tissue. These data show that siRNA can be used to modulate the ECM to improve collagen fiber formation and mechanical properties in tissue-engineered constructs, and that a transient decrease in aggrecan promotes the formation of a more robust fiber network.

## 1. Introduction

Menisci are fibrocartilage wedges located in the knee joint that facilitate load transmission, articulation, and shock absorption and provide a lubricated surface for joint stabilization [1]. The fibrous and cartilaginous structure of the tissue give it unique functional properties, making it resistant to compression, tension, and shear [1]. The biomechanical properties of the meniscus are associated with the structure and geometry of the tissue components [2]. The meniscus is made up primarily of water and collagen, which is organized into a complex fibrillar structure with proteoglycans, such as aggrecan, dispersed throughout this network [1,3,4]. The meniscal entheses anchor the meniscal body to the underlying bone. Entheses are composed of structural gradients, such as the complex and varying collagen orientation across the interface of bone to soft tissue, which play a prevalent role in the enthesis function [5,6]. The splayed collagen fiber orientation closer to the insertion into bone transitions to thicker, more oriented fibrils in the bulk meniscus tissue, which facilitate load distribution and are the primary drivers of the mechanical function of these tissues [5,7]. A persistent challenge in creating replacement options for damaged meniscal tissues is recapitulating this complex collagen architecture. When employing tissue engineering strategies for replacement options, it is important to consider the ECM components present and their structure during development of these tissues.

In the meniscus, both large aggregating proteoglycans and small leucine rich proteoglycans (SLRPs) contribute to the development and function of the tissue [3,8]. Glycosaminoglycans (GAGs), especially those associated with large aggregating proteoglycans, contribute to the compressive resistance of the meniscus by providing anionic charges needed to recruit water into the tissue [9,10,11]. It is well established that the interaction between proteoglycans and collagen influences the development of the collagen fiber network. SLRPs specifically are well known to influence collagen fibrillogenesis, but it is important to consider the roles of both large and small proteoglycans, and their associated GAG side chains, in the development of the tissue [12]. Prenatally, the organization of the collagen fiber network is established, and although SLRPS are present during the organizational stage, aggrecan content is minimal. While the fiber network matures postnatally, the foundational architecture of the collagen network is largely locked in at the early stages [13,14,15]. With age, proteoglycans are deposited in niches that are distinct from the fibers. Collectively, the fiber architecture and proteoglycan levels indicate that proteoglycan deposition and collagen organization happen at different time scales, with collagen organizing first. In fibrocartilage- and collagen-dense tissues, an excess deposition of proteoglycans during the developmental stages results in disorganized collagen architecture [16,17,18]. In vitro, fibrochondrocytes (FCCs) produce proteoglycans and associated GAG chains early in culture, which can inhibit collagen fiber formation during the maturation of tissue-engineered menisci [13,14,15,16,17,18,19]. Additionally, a negative correlation has been demonstrated between GAG deposition and fiber formation in tissue-engineered fibrocartilage [19,20]. Assessing the roles of large and small proteoglycans in the development of tissue-engineered constructs and modulating their presence at different stages of development could improve tissue structure and lead to more robust engineered fibrocartilage.

Previous work has used cell-seeded high-density collagen gels for engineered meniscus, allowing for the observation of large collagen fiber bundle formation [19,21,22,23]. Various techniques using biomechanical and biochemical stimulation have been used to harness cellular responses and perturb the extracellular matrix (ECM) to improve collagen fiber structure in tissue-engineered (TE) constructs [22,24]. Additionally, the idea of more closely mimicking native tissue development by regulating GAGs to improve collagen assembly has been addressed [25,26,27,28,29,30]. Our previous work studied the degradation of GAGs in the TE meniscus using chondroitinase ABC, which degrades chondroitin sulfate (CS) GAGs from both small and large proteoglycans [31]. Degradation of GAGs during maturation improved collagen fiber formation and mechanical properties in the TE meniscus. The GAG chains on aggrecan (>100) constitute the vast majority of GAGs in the meniscus, from which we have inferred that large aggregating proteoglycans are inhibiting fiber formation during in vitro development. Additionally, several studies from other groups suggest that SLRPs are beneficial to fiber formation and removal of their GAG chains can delay fibrillogenesis and lead to formation of thinner fibrils [32,33]. Thus, the removal of CS GAG chains from all proteoglycans may not be the optimal method for improving collagen fiber structure.

As such, there exists a need for methods to probe the specific contributions of small and large proteoglycans on collagen fiber formation in tissue-engineered fibrocartilage. Proteoglycan deposition in native tissue occurs in distinct temporal phases, with low aggrecan content early in development and higher content after fiber maturation [13,14,15]. Thus, it would be beneficial to both specifically and temporarily block deposition of proteoglycans early in development.

Small interfering RNA (siRNA) is a promising tool for specifically and temporally altering the production of ECM components by cells. siRNA delivery has been used to modulate the gene expression of many cell types [34]. siRNA has been used to control biosynthetic activity of articular chondrocytes; however, to our knowledge, only one previous study has used siRNA in meniscal fibrochondrocytes [35]. While the effects of siRNA are known to be transient, this temporary inhibition of production of specific ECM components may be highly desirable. This tool could be used to specifically and transiently inhibit the production of aggrecan, without affecting other critical ECM components and without long-term inhibition of subsequent beneficial proteoglycan deposition. The use of siRNA therapies to target and orchestrate matrix production for tissue engineering applications, however, has not been widely studied.

In this work, therefore, we examined the effects of transient inhibition of aggrecan via siRNA in fibrochondrocytes. The specific goals of this study were to (1) assess the effects of the inhibition of aggrecan production in both monolayer culture and in tissue-engineered meniscus constructs, (2) examine the effect of inhibition of aggrecan production on fiber formation and fiber alignment, and (3) assess the effect of siRNA treatment on bulk and microscale mechanical properties and their relationship to fiber organization in collagen gel-based tissue-engineered meniscus constructs.

## 2. Materials and Methods

### 2.1. Cell Isolation and siRNA Transfection

Fibrochondrocytes (FCCs) were harvested from 1- to 3-day-old menisci from a total of 37 bovine joints (Copper City Meats, Rome, NY, USA), as previously described [21]. Bovine joints were obtained from an FDA licensed abattoir, and because there were no live animals handled, this study was exempt from IACUC approval. Menisci were chopped and digested using 0.3 *w*/*v*% collagenase (Worthington Biochemical Corporation, Lakewood, NJ, USA) in Dulbecco’s modified Eagle medium (DMEM) solution with 100 µg/mL penicillin and 100 µg/mL streptomycin for 18 h. The digested tissue solution was then filtered through a 100 µm cell strainer and rinsed with phosphate-buffered saline (PBS) to remove collagenase. FCCs were then split into groups in preparation for transfection.

FCCs were either left untransfected or transfected using Lipofectamine RNAiMax (Thermo Fisher Scientific, Waltham, MA, USA catalog number 13778075). There were four groups used this study: (1) untransfected controls, (2) Lipofectamine transfection reagent controls (in monolayer and linear TE meniscal enthesis constructs), (3) siGLO (small interfering “GLO” RNA) Red siRNA controls, and (4) siACAN (small interfering aggrecan RNA) transfected constructs. siGLO Red is a fluorescent transfection indicator that localizes to the nucleus. siGLO Red does not interact with RISC or Cas9 nuclease, and therefore has no gene silencing properties. These siGLO Red indicators are Cy3-labeled with an absorbance/emission max of 547/563 nm.

Transfection of FCCs was based on previous studies conducted in chondrocytes [36]. For transfected groups, FCCs were either treated with Lipofectamine alone, or transfected with siACAN (Silencer Select, Thermo Fisher Scientific, Waltham, MA, USA siRNA ID ss1165) and Lipofectamine, or siGLO Red (Horizon Discovery, Boyertown, PA, USA catalog number D-001630-02) with Lipofectamine. siRNA and Lipofectamine complexes were prepared by diluting siRNA and Lipofectamine into DMEM without antibiotics or serum at a concentration of 10 pmol siRNA/10^6^ cells and 1.5 µg Lipofectamine/10^6^ cells. The siRNA/DMEM and Lipofectamine/DMEM mixtures were combined and briefly vortexed to form the complex. The siRNA/Lipofectamine complex was then combined with FCCs and briefly vortexed, and cells were incubated in suspension culture for 1 h at 37 °C. Then, cells were centrifuged to form a pellet and either immediately plated for 2D culture or seeded into collagen gels after transfection to avoid dedifferentiation [37].

### 2.2. D Culture of FCCs to Assess Suppression of Aggrecan Production via Biochemical Analysis

Following siRNA transfection, untransfected control, siGLO Red-transfected FCCs, and siACAN-transfected FCCs were placed into 24-well plates at 0.1 × 10^6^ cells/well. Cells were grown in 2D culture for 6 or 30 days. Media were collected every 3 days. Phase contrast and fluorescence images were taken of siGLO Red cells at each media collection. These images were used to calculate transfection efficiency by comparing the number of cells in the phase contrast image to the number of cells expressing fluorescence in the same field of view due to siGLO Red transfection.

To assess the GAGs produced by the FCCs throughout culture, sGAG and collagen content of the media in 2D was measured from each sample using a modified 1,9-dimethylmethylene blue (DMMB) assay and hydroxyproline assay, respectively [38,39]. For analysis of sGAGs, DMMB dye solution at a pH of 1.5 was added to media samples and analyzed at 525 nm. sGAG concentrations for experimental samples were calculated using a calibration curve of chondroitin sulfate GAGs. For assessment of hydroxyproline, samples were hydrolyzed before biochemical assessment and measured at 540 nm. Hydroxyproline samples were calculated using a calibration curve of hydroxy-4-proline.

### 2.3. Tissue-Engineered Construct Preparation

Type I collagen was extracted from the tendons of Sprague-Dawley rat tails (Pel-Freez Biologicals, Rogers, AZ, USA). Tendon bundles were removed from skinned rat tails and solubilized in 0.1 *w*/*v*% acetic acid and centrifuged, and the collagen supernatant was removed. The supernatant was lyophilized and reconstituted in 0.1 *w*/*v*% acetic acid at 30 mg/mL, as previously described [21,40].

To assess the feasibility of using siRNA in a 3D culture model and to then subsequently investigate how inhibition of aggrecan production affects fiber formation and mechanical properties in a collagen-based tissue-engineered construct, two model systems were used, as published previously: a disc and a linear geometry (Appendix A) [31,32,33,34,35,36,37,38,39,40,41]. Using a 3-way stopcock, a working solution of 1X PBS, 10X PBS, and 1M NaOH was mixed with reconstituted collagen solution to initiate gelation by bringing the pH of the solution to 7. This collagen gel was then mixed with the transfected or untransfected FCCs at a final concentration of 20 mg/mL of collagen gel and 25 × 10^6^ cells/mL [21,22]. For disc constructs, FCC-seeded gel was immediately injected between two glass plates set apart at 2 mm and allowed to gel before 8 mm punches were taken. For linear meniscus constructs, bone plugs were placed at either end of Tygon tubing^®^ and FCC-seeded gel was immediately injected into the center of the tubing. Gels were polymerized for 30 min at 37 °C [22,24,42]. These constructs contained either control cells with no siRNA, the Lipofectamine treatment with cells, siGLO with Lipofectamine in cells, or siACAN with Lipofectamine in cells. These cells were encapsulated in the gels using the mixing method described above.

Trabecular bone plugs used in the linear enthesis constructs were taken from the distal femurs of neonatal bovine joints (Copper City Meats, Rome, NY, USA) [42]. Bone plugs were extracted using a 6 mm diameter coring bit and washed using high-velocity streams of DI water to remove the bone marrow from pore spaces. Plugs were then washed with PBS and 0.1 *w*/*v*% ethylenediameinetetraacetic acid (EDTA) followed by a wash in hypotonic buffer (10 mM Trizma base, 0.1 *w*/*v*% EDTA). Plugs were then soaked in a detergent solution (10 mM Trizma base, 0.5 *w*/*v*% sodium dodecyl sulfate (SDS)) to remove cellular debris and punched to 4 mm diameter. Finally, plugs were washed with PBS with 100 µg/mL penicillin and 100 µg/mL streptomycin, lyophilized, and frozen. Before experimental use, plugs were soaked in ethanol, rinsed with PBS, and soaked in DMEM.

Once fabricated, disc constructs were placed into 24-well plates and cultured for 12 or 30 days. Enthesis constructs were mechanically clamped at the bone plugs to polylactic acid (PLA) molds to mimic native meniscus horn attachment [19]. All monolayer cells and constructs were cultured in 4500 mg/L glucose DMEM with 10% fetal bovine serum, 100 µg/mL penicillin, 100 µg/mL streptomycin, 50 µg/mL ascorbic acid, 0.4 mM L-proline, and 0.1 mM nonessential amino acids at 37 °C and 5% CO_2_. High-glucose media not only increases overall metabolism but also increases the production of proteoglycans in this system [24]. This media formulation makes this system well suited to probing questions about how suppressing large amounts of proteoglycan production affects the organization of other ECM components. Culture media were collected and changed every 3 days, and constructs were kept in culture for 30 days.

### 2.4. Biochemical Analysis of Tissue-Engineered Constructs

After a 12- or 30-day culture period, disc or linear constructs were weighed to obtain wet weight and sectioned for biochemical analysis. They were then lyophilized, weighed again to obtain dry weight, and digested in 1.25 mg/mL papain solution (Sigma-Aldrich, St. Louis, MO, USA). To assess the GAGs produced by the FCCs throughout culture, sGAG content of the media in 3D culture and constructs were measured from each sample using a modified 1,9-dimethylmethylene blue (DMMB) assay [38]. DNA and collagen contents were also measured using Hoescht DNA assay and hydroxyproline assays, respectively [39,43]. For construct samples, sGAG, DNA, and hydroxyproline values were normalized to wet weight.

### 2.5. Microscopy Analysis of Fiber Formation

To assess the effects of inhibition of aggrecan production on the collagen structure at the bulk tissue scale, microscale, and nanoscale, multiple microscopy methods were applied. Picrosirius red histology imaged under polarized light was employed to study the tissue as a whole. The birefringence of the collagen as seen in polarized light micrographs of picrosirius red can be correlated with fiber size and orientation [44]. Second harmonic generation (SHG) imaging was used to assess the structure at the fiber level and scanning electron microscopy (SEM) was used for the visualization of the fibril structure. SHG and SEM allows for these tissues to be imaged at a smaller scale, allowing us to glean information on fiber and fibril orientation and diameter.

#### 2.5.1. Microscopy Analysis of Fiber Formation—Histological Analysis

Linear enthesis constructs (*n* = 4) were taken after the 30-day culture period, fixed in 10 *v*/*v*% of buffered formalin for 48 h, and stored in 70 *v*/*v*% ethanol. Samples were then embedded in paraffin and sectioned for histological analysis. Sections were stained using picrosirius red and Weigert’s hematoxylin as a counter stain before being imaged under polarized light. Histology slides were examined and imaged using a Nikon Eclipse TE2000-S microscope (Nikon Instruments, Melville, NY, USA) with a SPOT RT camera (Diagnostic Instruments, Sterling Heights, MI, USA).

#### 2.5.2. Microscopy Analysis of Fiber Formation—Second Harmonic Generation Imaging

Disc constructs (*n* = 3) from each group were used for SHG imaging. An LSM 880 confocal/multiphoton inverted microscope with 10×/1.2 N.A. C-Apochromat water immersion objective using SHG and autofluorescence were used to observe circumferential collagen fibers [22,24]. Cellular autofluorescence was imaged between 495 and 580 nm, and collagen fiber reflectance was measured between 437 and 464 nm [24]. A total of 4 SHG images were taken per sample to be analyzed and were averaged by sample. The fiber alignment index was calculated using custom MATLAB code, as described previously [22]. This code uses a series of 2-dimensional fast Fourier transforms (FFTs) to determine the maximum degree of alignment of the fibrils in each SHG image. The alignment index can then be calculated using the intensity ±20° from the maximum degree of alignment found previously. Alignment index values fall between 1 (unaligned) and 4.5 (completely aligned).

#### 2.5.3. Microscopy Analysis of Fiber Formation—Scanning Electron Microscopy

After 30 days in culture, 3–4 constructs from each group were fixed in 2% glutaraldehyde in HEPES buffer for 24 h. Samples were then washed with HEPES [×2] for 10 min each and subsequently dried with a series of dilution of water into ethanol (25, 50, 70, 90, and then 100% [×2]) for 10 min each. After soaking in 100% ethanol for 24 h, samples were dried using critical point CO_2_ until complete ethanol removal. Samples were fixed to 18 mm aluminum specimen mounts using double-sided adhesive and silver conductive paint [45]. Prior to imaging, samples were sputter-coated with gold/palladium alloy for 15 s at a target current of 20 mA [45]. Prepared construct samples were imaged on a Tescan Mira3 field emission SEM (Cranberry, Township, PA, USA) [45]. Collagen fibril diameters were measured manually using ImageJ (Version 2.9.0/1.51W). Images were divided into a 4 × 4 grid and the diameters of fibrils were measured until the average remained constant (after measurement of ~100 fibrils per micrograph).

### 2.6. Macroscale Mechanical Analysis Using Bulk Tensile Tests to Failure

Tensile tests were performed on the tissue-engineered linear enthesis constructs on an Enduratec Electroforce 5500 System (Bose, Eden Prairie, MN, USA), as previously described [42]. Full-length linear meniscus constructs were clamped at the bone attachments to ensure consistent testing conditions and secure grip. To simulate quasi-static loading, these samples were subjected to a tensile pull to failure at a constant displacement rate of 0.15 mm/s, as described in previous studies [42]. The length of each construct was determined as the measured distance between the two bone/collagen interfaces. Ultimate load, Young’s modulus, toughness, and ultimate tensile strength were calculated from measured stress and strain curves [42].

### 2.7. Microscale Mechanical Analysis Using Confocal Fluorescence Elastography

To investigate the local microscale, strain distributions, strain analysis was performed on linear meniscal enthesis samples (*n* = 4–5), as previously described [46]. Samples were embedded in OCT, longitudinally bisected, and then stained with 5-(4,6-Dichlorotriazinyl) aminofluorescein (5-DTAF, ex/em 492/516 nm, Invitrogen, Waltham, MA, USA), a general protein stain, for 30 min. After staining, samples were rinsed three times with PBS and then were mounted onto a manually actuated tissue deformation imaging stage (TDIS). One bone plug was cut off, the other bone plug was glued to a fixed back plate, and the soft tissue collagen portion of the construct was glued to a loading plate. After being fixed to the TDIS, samples were submerged in PBS throughout testing. The TDIS was placed on an inverted Zeiss LSM 710 confocal microscope and a 15% strain was applied to each sample in incremental 1% steps. Five minutes of stress relaxation was allowed to pass before images were taken at each step using a 5× objective and a 488 nm laser. Images were analyzed using MATLAB NCorr (Matlab 9.5 R2018b), an open-source digital imaging software package, to obtain spatial strain maps to quantify strains. Maximum local strains were calculated as the average of local strains in the 95th percentile throughout the tissue. Histograms were created for each sample to visualize the frequency of strains. To assess the multimodal distribution of the strain data, two Gaussian curves were fitted to each histogram and the means of each curve were assessed, as well as the proportion of data accounted for by each curve. This analysis allowed us to compare the means of each curve between groups.

### 2.8. Statistical Analysis

Values in this study are reported as mean ± standard deviation and *p* < 0.05 was considered significant. Data were analyzed using one-way analysis of variance with Tukey’s honestly significant difference for post hoc analysis. Mean strain values were log transformed prior to running the one-way ANOVA and post hoc analysis. A Pearson correlation coefficient was computed to assess the linear relationship between max local strain and fibril diameter. Statistical analysis was performed using R Statistical Software (Version 2022.07.01+554, R Foundation for statistical computing, Vienna, Austria) and GraphPad Prism (Version 10.1.1 (323), GraphPad Prism Software INC., San Diego, CA, USA).

## 3. Results

### 3.1. siRNA Transfection of Fibrochondrocytes (FCCs) in 2D Culture

To demonstrate the feasibility of controlling aggrecan synthesis using siRNA, we transfected siRNA into FCCs and maintained them in monolayer culture. ACAN (aggrecan) siRNA was used to inhibit aggrecan production, siGLO Red was used as an siRNA control, Lipofectamine was used as a transfection control, and untransfected cells were used as controls. At 3 days of culture, transfection efficiency was 59 ± 13% (Appendix A), as determined by comparison of phase contrast and fluorescence images of the siGLO Red-transfected cells (Figure 1A), and comparable with other studies transfecting chondrocytes [36,47]. Live/Dead staining showed that cells with Lipofectamine or siACAN had a lower viability, at 87.1 ± 4.0% (*p* = 0.01) and 88.3 ± 1.8% (*p* = 0.03), respectively, on day 3 compared to controls (93.9 ± 1.2%). The viability increased by day 6, with Lipofectamine cells at 96.5 ± 3.3% (*p* = 0.25), siACAN cells at 96.4 ± 1.1% (*p* = 0.22), and control cells at 99.0 ± 0.7%. All cells started culture with either rounded or elongated shapes, transitioning to elongated phenotypes after 6 days, as expected of fibrochondrocytes grown in 2D [37]. Aggrecan production by FCCs in monolayer culture was dramatically reduced out to 6 days (Figure 1B) in those cells treated with siACAN as compared with the controls, and collagen production, as assessed by hydroxyproline content, was not affected (Figure 1C). This experiment demonstrated the feasibility of using siACAN to regulate proteoglycan production by meniscal FCCs, without affecting production of the main ECM constituent, collagen.

### 3.2. siRNA Transfection of FCCs in 3D Culture—Disc Constructs

TE meniscal disc constructs were used to assess the transient effects of the siRNA, as well as to see if siRNA-transfected FCCs perform similarly in 3D culture. siACAN-transfected, siGLO Red siRNA control, and untransfected control FCCs were used. The effects of siRNA treatment were assessed at both early and late stages of culture. At 3 and 9 days in monolayer culture, the siGLO Red fluorescence signal remained strong. After 12 days in monolayer culture, siGLO Red fluorescence signal decreased, with little signal found at day 30 (Figure 2A). Collagen gel constructs with FCCs transfected with ACAN siRNA, siGLO Red, Lipofectamine, or transfected controls were generated, punched into 8 mm discs, and cultured before assessment of GAG and hydroxyproline content at 12 and 30 days of culture, as well as fiber structure at 30 days. At 12 days, we saw a 67% (*p* = 0.02) suppression of GAG accumulation due to siACAN, but by 30 days there was no significant difference between any of the groups (35%, *p* = 0.17) (Figure 2B), consistent with the signal found in monolayer culture. Second harmonic generation (SHG) microscopy was used to characterize and visualize the collagen fiber morphologies after 30 days of culture. Notably, even the temporary suppression of aggrecan synthesis had a profound effect on fiber formation of these constructs after 30 days, where fiber diameter increased by 29% and 75% compared to siGLO and untransfected control, respectively (Figure 2D,E).

### 3.3. Linear Meniscal Enthesis Constructs

A linear meniscal enthesis system was used to further characterize the effects of the inhibition of aggrecan production in FCC-seeded tissue-engineered constructs. These constructs consist of a cylinder of collagen gel with decellularized trabecular bone plugs on either end. During culture, the bone plugs are clamped to mechanically anchor the constructs, further facilitating fiber maturation, and allowing for the assessment of tensile properties [31]. These linearly aligned enthesis constructs allow for the observation of fiber formation at the interface of the bone plug and soft gel and the bulk soft tissue and for tensile testing. Fiber organization of these constructs was assessed at the bulk tissue level using histology, as well as at a subcellular scale using SEM imaging. Tensile tests were performed on these tissues to assess how fiber and fibril organization affects the mechanical properties of these tissues.

#### 3.3.1. Assessing Fiber and Fibril Organization in Constructs via Polarized Light Microscopy and Scanning Electron Microscopy

The organization of collagen fiber networks at the whole tissue level was visualized using polarized light microscopy of picrosirius red-stained linear meniscal enthesis constructs. Untransfected control, Lipofectamine, and siGLO constructs displayed little birefringent signal, indicating weak fiber formation, especially seen in control and siGLO constructs (Figure 3). The siACAN group showed a slight increase in birefringence, as indicated by increased yellow and green signals, and seen most prominently on the edges of the construct (Figure 3, white arrows). Although there does not appear to be consistent or well-established fiber orientation, the siACAN group appears have fibers parallel to the interface that turn perpendicular in the bulk tissue.

To assess the collagen organization at a more local level, we looked at the fibril alignment and fibril diameter of collagen in these tissues using SEM. siACAN-transfected constructs displayed an increased collagen fibril diameter compared to all three control groups. siACAN constructs showed a significant increase (*p* < 0.01) in fibril diameter; however, there were no differences in fiber alignment between all four groups, consistent with the orientation observed at the bulk level in histology (Figure 4B,C).

#### 3.3.2. Bulk and Local Tensile Properties

To investigate how the increase in fibril diameter affects the mechanics of the tissue-engineered meniscal entheses, we performed bulk and local tensile testing. Assessment of the bulk tensile properties of the constructs revealed no statistical difference between the groups (Figure 5). The ultimate load showed similar means for siGLO, Lipofectamine, and siACAN groups, and the ultimate load for siACAN was 63% higher (*p* = 0.45) than that for the control (Figure 5A). Toughness displayed a slight increasing trend going from the untransfected control to siACAN. The toughness of siACAN was increased by 64% (*p* = 0.3) compared to the control (Figure 5B). A similar trend was seen in Young’s modulus and UTS, with 125% (*p* = 0.37) and 117% (*p* = 0.46) increases, respectively, in siACAN compared to control constructs (Figure 5C,D). siGLO and siACAN constructs were comparable in all bulk mechanical properties. Notably, the spread of the data was large and there was a slight increasing trend in the average, with the untransfected control having the lowest values, followed by Lipofectamine, siGLO, and finally siACAN.

Noting the differences in mean values between control and siACAN groups, and the associated large spread of the data, we were interested in assessing how the change in fiber structure would affect the mechanical properties at a microscale level, since this length scale is where we observed the biggest changes in structure. Using confocal elastography, strain maps obtained at 15% applied strain show striking differences in the distribution and magnitude of strains experienced by the tissues (Figure 6A). Strain maps of the linear constructs for all groups showed concentrations of strain near the bone/collagen interface. These strain maps showed that in all control conditions, there was close to a 30% or greater strain near this interface. In the siACAN group, these interfaces experienced closer to 20% strain. To visualize this difference, we looked at the maximum local strains as quantified by the 95th percentile strains experienced by the tissue. siACAN-transfected constructs showed a decreased 95th percentile maximum local strain, which was correlated with increased fibril diameter, as shown by SEM, in this group (Figure 6B,C). Untransfected control, Lipofectamine control, and siGLO control all showed peaks of strain near the interface of the bone and soft tissue region in these constructs (Figure 6A).

Strain histograms were plotted for each sample and two Gaussian curves were fitted to the histograms (Appendix A). The means for each Gaussian curve, as well as the proportion of data under each Gaussian curve, were plotted (Figure 7 and Appendix A). The mean strains found using these curves show distributions of strains that are nicely broken up into high- and low-strain regions. The means of the first Gaussian curve remained at 10% strain and lower, matching the strain found in the bulk tissue, as seen in the strain maps (Figure 6A). The mean strains of the first curve for the siACAN group were 39% (*p* = 0.78), 64% (*p* = 0.22), and 73% (*p* = 0.11) lower than control, Lipofectamine, and siGLO, respectively. The means of the second curve were more closely matched to those found at the interface of the bone plug and soft tissue in the samples (Figure 6A). These strains in the siACAN group were 35% (*p* = 0.44), 51% (*p* = 0.043), and 55% (*p* = 0.025), lower than control, Lipofectamine, and siGLO, respectively. These data show that there is a clear effect of siACAN at the bone/collagen interface, increasing the ability of these tissues to effectively distribute strains.

## 4. Discussion

The objective of this study was to investigate whether specific inhibition of aggrecan synthesis via siRNA treatment during TE meniscus construct maturation would increase collagen fiber diameter and alignment, to be better suited to distribute strains at the bone/collagen interface and if these fiber changes improved tensile properties at the bulk scale and microscale. Our previous work showed the degradation of chondroitin sulfate GAG chains from all proteoglycans improved fiber diameter and alignment and subsequent failure properties in this system [31]. We hypothesized, however, that the GAG chains on aggrecan are likely the main contributors to any inhibitory effect on collagen fiber formation. We also hypothesized that SLRPs and their GAG chains play an important role in fibrillogenesis. Therefore, we proposed that a more specific method of large proteoglycan removal would further improve the fiber architecture. This improvement would increase the ability of the tissue to distribute strains, particularly at the interface of the bone and soft tissue, and improve the tensile mechanics of the tissue. We combined our previously established collagen-based TE meniscus model and mechanical anchoring of these constructs with specific inhibition of aggrecan production to try to further increase fiber diameter and alignment. We found that even temporary suppression of aggrecan production significantly improved the collagen fibril diameter, whereas bulk tissue organization remained largely unchanged. Improved fiber architecture reduced strain concentrations at the bone/collagen interface in the linear enthesis model.

Since its discovery over 20 years ago, siRNA delivery has been used to modulate the gene expression of many cells and has been used to remodel the extracellular matrix for the purpose of changing the matrix properties [34,48]. The use of siRNA therapies to target and orchestrate matrix production for tissue engineering applications has not been widely studied. In this study, we used siRNA to specifically and temporarily inhibit the production of aggrecan by neonatal bovine fibrochondrocytes. The transfection efficiency of these cells was 59%, which is comparable to other studies using siRNA in chondrocytes [36,47]. The viability of Lipofectamine- and siACAN-treated cells was initially slightly lower than that of controls, but by day 6, cells were nearly identical in viability compared to controls (*p* = 0.25 (Lipofectamine) and *p* = 0.22 (siACAN)). siRNA knockdown of aggrecan production was successful in 2D culture (Figure 1B). Although the standard deviations appear large, the small scale of GAG production measurements from monolayer cell culture magnifies their relative impact, exaggerating the variation. Knockdown was also successful in 3D tissue culture, allowing for the tuning of ECM production to orchestrate construct development (Figure 2B).

Successful inhibition of aggrecan production by siRNA remained effective for about 12 days in culture, as indicated by loss of siGLO signal in 2D culture (Figure 2A) and as assessed by GAG accumulation at 12 and 30 days in TE disc constructs (Figure 2B–E). While large aggregating proteoglycans such as aggrecan are important to the compressive function of the developed meniscus, the temporal nature of their deposition during tissue maturation is key in allowing for the complex organization of the collagen matrix to occur. In the native meniscus, aggrecan content is minimal in early development while the collagen network is established [13,14,15]. As the tissue matures, proteoglycans are deposited in distinct niches, notably after a collagen network architecture has been formed. Additionally, it has been shown in fibrocartilage, both in vivo and in vitro, that excess proteoglycan deposition during development can inhibit the formation of an organized collagen fiber network [13,16,19]. Notably, in the TE disc constructs in this study, even a suppression of aggrecan production up to 12 days facilitated increased fiber diameter in transfected constructs grown out to 30 days in culture.

The meniscus develops and functions under a variety of loading conditions. Our lab has established mechanical anchoring techniques to be applied in culture to facilitate the maturation of collagen fibers in TE constructs as they develop under load [22]. In addition, the use of a linear meniscal enthesis system allows for the assessment of tensile property changes due to altered collagen fibril structure [42]. In these collagen-based gels, fibrochondrocytes remodel and arrange both the existing and FCC-produced collagen, which is minimal compared to the amount of initial collagen in the gel (20 mg/mL). This system allows for the retention of collagen and cells, and provides favorable sites for the cells to attach and remodel the matrix [21]. Polarized light micrographs of picrosirius red histology showed that the siACAN group showed some parallel fibers close to the interface and non-continuous perpendicular fibers in the bulk tissue (Figure 3). Overall, the siACAN group showed the most yellow birefringence, indicating more organized collagen network compared to the three controls. The three control groups displayed a dim and greener signal, indicating less fiber organization. Though the siACAN group showed improvement over the three controls, there was no evidence of organization of large collagen fiber bundles over bulk millimeter-scale distances, as seen in native tissue. On the smaller scale, after 30 days of culture, siACAN-transfected linear meniscal constructs displayed a fibril diameter more than double that of the three control groups, as determined by quantitative analysis of SEM images (Figure 4). Increased fibril bundling was seen in siACAN constructs but fibril alignment remained unchanged between all four groups. At the tens of microns length scale, we see an increased effect of siACAN on collagen network structures. Fibril diameters were significantly increased, as shown by SEM, but bulk scale organization was largely unaffected, as indicated by histology.

Organized collagen networks are essential for the mechanical function of the meniscus, in particular for distributing strains at the bone/meniscus interface, and it is important that this native structure is mimicked in TE constructs. Our group has shown that regulation of proteoglycan production by varying glucose concentrations enhances fiber formation, with decrease in GAG production occurring at 500 mg/L of glucose, corresponding to peak increases in fiber diameter [24]. Additionally, our lab has shown that by degrading the chondroitin sulfate GAG chains in these collagen-based TE constructs, fiber organization and alignment were improved [31]. Removal of these GAGs improved tensile properties, and the compressive strength of the tissue was not compromised. Other studies using chondroitinase ABC as a tool to remove GAGs from tissues have also shown improvements in collagen network formation in TE meniscus and cartilage [25,26,28]. Here, we demonstrated that fibril diameters were greatly improved by combining a mechanical boundary constraint with the specific inhibition of aggrecan production.

Assessment of tensile properties is essential when designing a TE meniscus replacement. Because we are seeing changes to the tissue at the fibril level, it is important that the tensile properties at the macroscale and the microscale are considered. Although no statistically significant changes in the bulk mechanical properties were observed between the three control groups and the siACAN group due to the variability of the data, the averages of the Lipofectamine, siGLO, and siACAN groups showed a slight upward trend compared to the control. Notably, siACAN constructs demonstrated a 63% higher ultimate load than the control (*p* = 0.45). In addition, the average toughness, Young’s modulus, and UTS of these three groups also displayed a slight increasing trend compared to the untransfected control. Toughness, Young’s modulus, and UTS were increased in siACAN constructs by 64% (*p* = 0.3), 125% (*p* = 0.37), and 117% (*p* = 0.46), respectively, compared to the untransfected control. Within the siACAN group, the bulk mechanical data suggest the presence of two distinct subpopulations: one resembling the control group and another exhibiting enhanced mechanical properties. While the overlap of these subpopulations results in overall mechanical properties that are not significantly different to controls, the presence of a subset with increased mechanical properties is noteworthy. These mechanical data are still orders of magnitude below that of native human meniscus tissue, with Young’s moduli reaching 59 MPa [49]. Therefore, continued work must be done to improve mechanics for use as tissue-engineered replacements.

These tissue-engineered meniscus constructs are complex tissues to test due to their heterogeneity and anisotropy [50]. The picrosirius red histology micrographs (Figure 3) show that the tissues are not homogenous. There exist distinct structures on the edges of the constructs that are different from the bulk gel, and it is clear at this bulk scale that even the midsubstance qualitatively seems to vary in orientation of the fibers, their aggregation, and their presence [51]. When evaluating the bulk tensile properties of these gels, we may not fully account for their heterogeneity [52]. This limitation could explain the lack of statistical significance in the bulk mechanical data between groups (Figure 5), despite the presence of more highly organized fiber bundles in the siACAN-treated groups (Figure 4). Previous studies have employed microscale approaches to investigate gradients and heterogeneity in tissue mechanical properties [41,53,54]. Such microscale assessments enable the exploration of how microstructural changes in fiber organization impact mechanical behavior, even when the tissue-level fiber structure does not substantially alter the bulk properties of the tissue. Therefore, we wanted to assess how these microstructural changes contributed to the micromechanics of these tissues.

To assess the micromechanics of these tissues, we used a tissue deformation imaging stage mounted on a confocal microscope, which allowed us to visualize and map how the strain was distributed across the tissues. At 15% applied strain, strain maps of all four groups show striking differences between controls and the siACAN constructs in terms of both distribution and magnitude of strains experienced by the tissues. In control groups, a large concentration of strain was seen near the bone and soft tissue interface, consistent with previous studies [46]. The average local strains experienced by the control, Lipofectamine, and siGLO samples were 1.89-, 1.88-, and 2.07-fold higher than the applied strain of 15%, respectively. siACAN constructs experienced local strains only 1.12-fold higher than the applied strain of 15%. Correlating the fibril diameter found using SEM images and the maximum local strain revealed a significant negative correlation, showing that increased fibril diameter led to decreased max local strains experienced in the tissue. Therefore, the increased diameter of the collagen fibrils in the siACAN group could explain why there was increased ability of these tissues to dissipate strains at the bone/collagen interface, as increased fiber diameters have been correlated with increased tensile strength [55]. The two Gaussian curves describe different portions of the construct. The first curve fits the properties of the bulk tissue, as compared to the strain map (Figure 6A and Figure 7). This bulk area experiences less concentrated strains and the strains are at lower values. The second curves fit the strains that the tissues experience at the interface of the bone and soft tissue. These strains, especially for control, Lipofectamine, and siGLO constructs, are more concentrated and have higher values. The strains experienced by the siACAN constructs at the interface are lower than those in control (*p* = 0.44), Lipofectamine (*p* = 0.043), and siGLO groups (*p* = 0.025), indicating that the siACAN constructs are able to distribute these strains more effectively.

siACAN-treated constructs exhibited enhanced collagen organization at the fibril level, resulting in thicker fibrils and improved local structure. The fiber structure at the bulk level, however, was only slightly improved compared to controls. While the larger fibril diameter contributed to increased ability of siACAN constructs to locally dissipate strains at the bone/collagen interface, bulk tissue mechanics remained similar to controls. The second Gaussian curves fitted to the strain histograms further describe that the siACAN group is more effective at distributing strains at the interface, which was correlated to increased local fibril diameter. These results suggests that while inhibition of aggrecan production via siRNA fosters the development of thicker fibrils and enhances local structure, it does not impact the continuity or organization of these fibrils into larger fiber bundles across the bulk tissue. This interpretation is additionally supported with the picrosirius red histology micrographs, which showed some fibers parallel to the interface in the siACAN group and non-continuous perpendicular fibers in the bulk tissue.

One of the challenges of siRNA-based approaches is determining that the target siRNA treatment and the vehicle used for siRNA delivery do not have any off-target effects. While it is difficult to comprehensively quantify the off-target effects of each siRNA in a particular cell line, there is a strong chance that there are off-target genes that are affected. In this system, we have looked at fiber diameter in both disc and linear constructs, where fiber diameter is significantly higher in the siACAN groups compared to controls (Figure 2E and Figure 4B). In the disc constructs, siGLO does have a nominally elevated fiber diameter compared to untransfected control samples. This may mean that the vehicle (Lipofectamine) has some effect, likely some level of cytotoxicity, which could be altering fiber diameter. Notably, if there is some early level of cytotoxicity due to Lipofectamine, this would decrease proteoglycan production, which we think is a major driver of collagen fiber assembly. This is consistent with the slight, but not statistically significant, decrease in proteoglycan production in the disc at day 12 (Figure 2B). All of these data collectively point to the idea that suppressing proteoglycan production enhances fiber formation. In this study, we assessed the off-target issue by looking at collagen production. Collagen is the main secretory product of meniscal fibrochondrocytes, and hydroxyproline production and maintenance were not affected by siRNA/Lipofectamine delivery (Figure 1C and Figure 2C). While this does not eliminate the possibility of other off-target genes being affected, this does show that the production of collagen, the main protein we are interested in, remained unchanged, and that aggrecan was inhibited, as determined via assessment of GAG production. Additionally, we looked at whether the siRNA control, siGLO, or the Lipofectamine vehicle control had any unwanted effects. We saw that there was a nominal but non-statistical decrease in the DNA of the siGLO disc constructs compared to untransfected controls (Appendix A). Notably, the siGLO and Lipofectamine groups also had nominal changes in bulk mechanics (Figure 5) and fiber diameter in the disc constructs (Figure 2E). Decreasing cell number would also decrease aggrecan production and it is possible that some of the non-statistically significant effects in bulk mechanics or fiber diameter are due to this non-statistical decrease in DNA. However, it is clear that siACAN inhibition resulted in significant changes in fibril structure and local mechanics.

The cells transfected in this study were from neonatal tissues and further exploration of transfection of fibrochondrocytes at differing stages of age and development may be necessary for future applications of this work. Additionally, siGLO Red was used as both a transfection indicator and a negative control, as it does not silence any genes but is an siRNA that is transfected via cationic lipids (Lipofectamine) [56]. It may be useful in future experiments to include a random scrambled siRNA; however, siGLO serves as a negative control in this study, having no known target in the meniscal fibrochondrocytes used in this study. Additionally, siRNA can have unintentional off-target effects when the suppression of unintended mRNAs with partially complementary sequences occurs [57]. While RNA-seq or RT-PCR could be used to determine the extent of these off-target effects, there were no appreciable effects, especially on the production of collagen (Figure 2C) in these constructs. The goal of this study was to understand whether siRNA could be used to knock down aggrecan by meniscal fibrochondrocytes in an in vitro collagen gel model, and thus influence collagen fiber formation and change mechanics. Any off-target effects that may be present did not negatively impact the production of collagen, the most prevalently produced protein by FCCs, nor did it hinder the goals of this study of assessing how aggrecan knockdown influences collagen fiber formation.

## 5. Conclusions

In conclusion, siACAN treatment inhibited aggrecan production in 3D culture without affecting the production of collagen, demonstrating the feasibility of this tool to regulate ECM production in TE meniscus constructs. SEM images show that specific inhibition of aggrecan production produced a striking difference in fiber formation, specifically in fibril diameter, where we saw more than a doubling in siACAN constructs compared to controls. Bulk tensile mechanical properties showed nominal but not statistically different differences due to the wide data spread between siACAN-treated constructs compared to controls. At the local level, however, the maximum local strain experienced by siACAN-transfected constructs decreased by 42% compared to control groups combined. This decrease in local strains experienced by the siACAN group is due to the increased collagen fibril diameters found in these constructs, as indicated by the correlation (Figure 6C). Distinct architectural and functional changes occur in constructs with suppressed aggrecan production, and these data support that a transient decrease in aggrecan promotes the formation of a more robust fiber network, especially a larger fiber diameter. This robust fiber network may allow fibrils to more effectively distribute strains at the local level. Further exploration of proteoglycan content in developing TE meniscus constructs is needed to optimize collagen fiber formation and subsequent mechanical properties. This study demonstrates the feasibility of using siRNA as a tool to engineer the ECM of TE constructs to enhance their mechanical properties and tailor their microstructure, with the ultimate goal of improving these tissues for use as meniscus replacements.

## Figures and Tables

**Figure 1 bioengineering-11-01308-f001:**
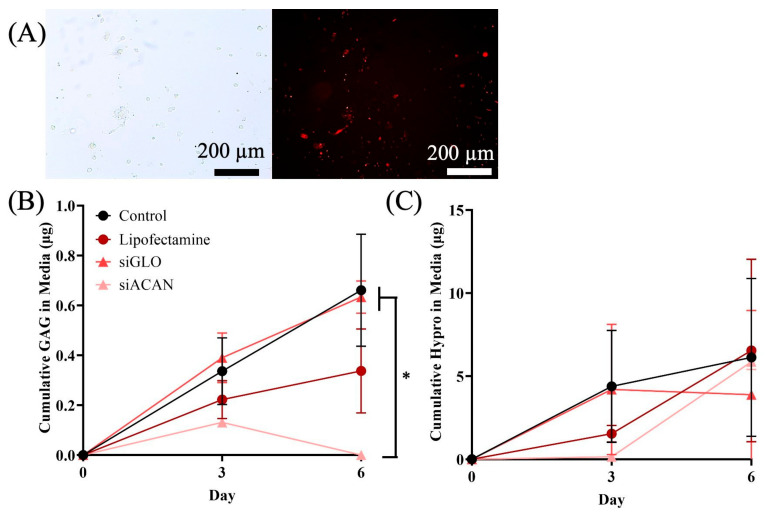
(**A**) Phase contrast (left) and fluorescence (right) images of siGLO monolayer cell culture at day 3. (Scale bars = 200 µm) (**B**) GAG production of FCCs transfected with ACAN siRNA (light pink triangles), siGLO (dark pink triangles), Lipofectamine (red circles), or untransfected controls (black circles) in media over 6 days in 2D culture (**C**) Hydroxyproline production of FCCs transfected with ACAN siRNA, siGLO, Lipofectamine, or untransfected controls in media over 6 days in 2D culture. Analyzed using 1-way ANOVA with Tukey’s multiple comparisons test (* = *p* < 0.05).

**Figure 2 bioengineering-11-01308-f002:**
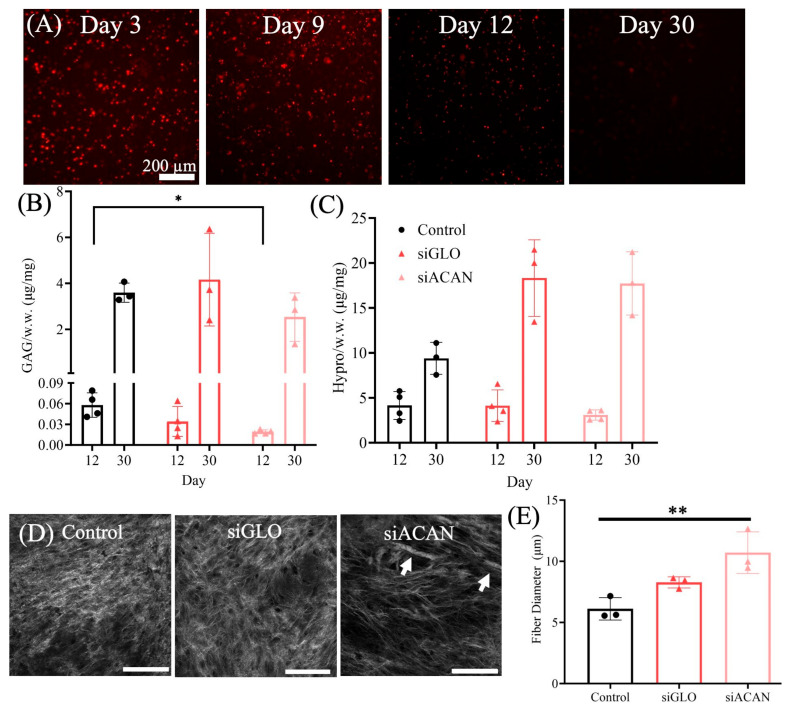
(**A**) Fluorescence images of siGLO 2D monolayer culture at 3, 9, 12, and 30 days of culture. (**B**) GAG production after 12 or 30 days in 3D tissue culture (*n* = 3–4). (**C**) Hydroxyproline production after 12 or 30 days in 3D tissue culture (*n* = 3–4) (**D**) Representative SHG images of constructs cultured for 30 days (scale bar = 100 µm). White arrows indicate fibers with larger diameters. (**E**) Quantitative analysis of fiber diameter in meniscus constructs (*n* = 3) at 30 days of culture. Analyzed using 1-way ANOVA with Tukey’s multiple comparisons test (* = *p* < 0.05, ** = *p* < 0.01).

**Figure 3 bioengineering-11-01308-f003:**
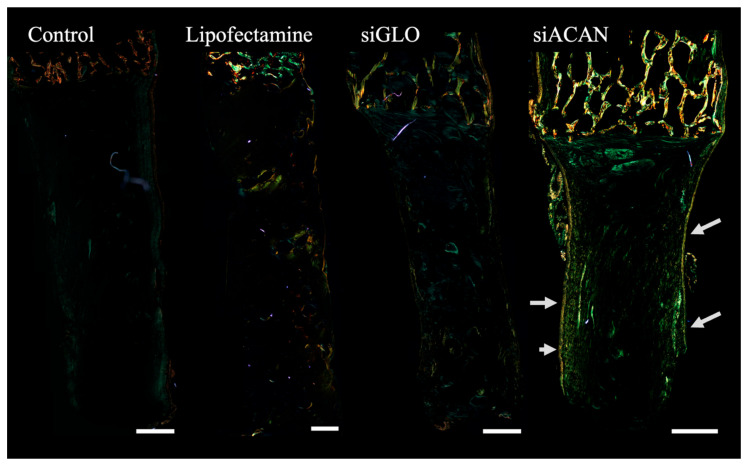
Polarized light micrographs stained with picrosirius red. Bone plugs can be seen on top of the images, displaying a more orange and yellow coloring. Soft gel can be seen below the bone plugs with birefringence of collagen fibers displayed in green and yellow (white arrows indicate increased birefringence on edges of siACAN construct). More oriented and thicker fibers display increased birefringence in yellow and slightly orange colors. (All scale bars = 1 mm).

**Figure 4 bioengineering-11-01308-f004:**
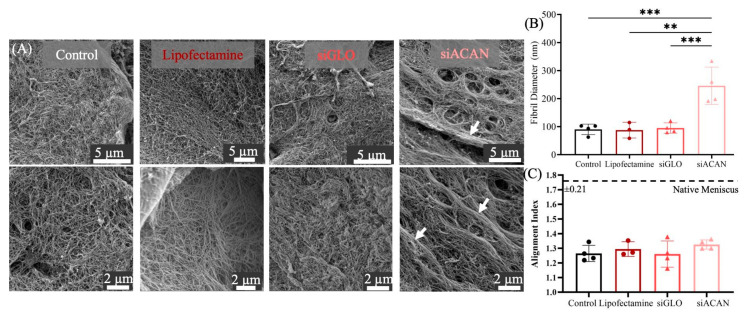
(**A**) Representative SEM images of critical point dried linear meniscal enthesis constructs cultured for 30 days. White arrows indicate regions containing thicker collagen fibril bundles. (**B**) Quantitative analysis of fiber diameter in linear meniscal enthesis constructs from SEM images (*n* = 3–4) at 30 days of culture (** = *p* < 0.01, *** = *p* < 0.001). (**C**) Quantitative analysis of fiber alignment in linear meniscal enthesis constructs from SEM images (*n* = 3–4) Dashed line indicates average alignment index found in native menisci (1.76 ± 0.21) (** = *p* < 0.01, *** = *p* < 0.001).

**Figure 5 bioengineering-11-01308-f005:**
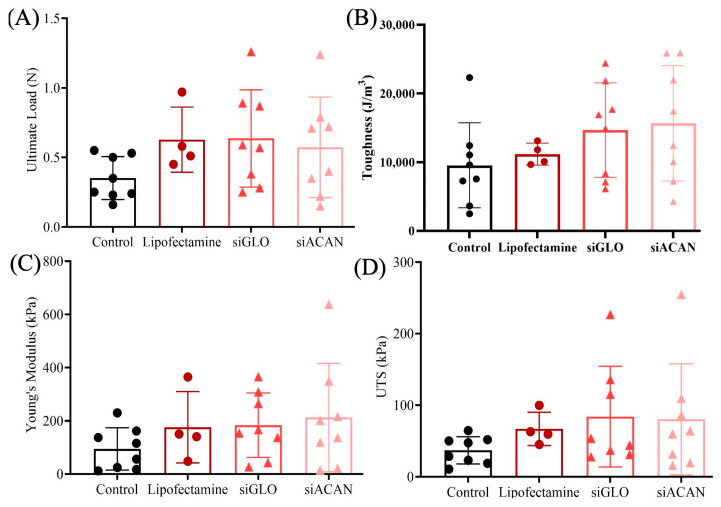
Tensile properties of meniscal enthesis constructs (*n* = 4–8). (**A**) Ultimate load. (**B**) Toughness. (**C**) Young’s modulus. (**D**) Ultimate tensile strength. Statistical error bars are ± standard deviation. Analyzed using 1-way ANOVA with Tukey’s multiple comparisons test.

**Figure 6 bioengineering-11-01308-f006:**
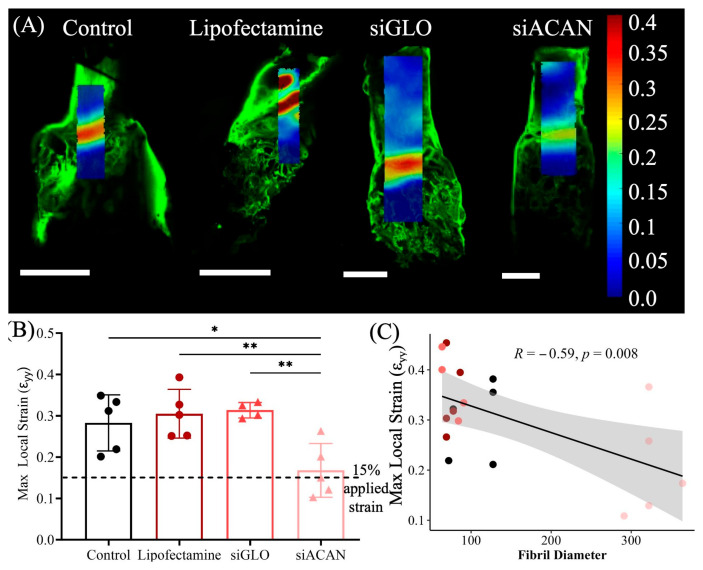
(**A**) Representative local strain (ε_yy_) maps of control, Lipofectamine-treated, siGLO-treated, or siACAN-treated tissue-engineered meniscal enthesis constructs at 15% bulk strain. Scale bars = 2 mm. Fluorescent confocal images were taken at every 1% strain. Local strains were calculated based on displacements tracked within a selected region of interested using MATLAB’s NCorr program. Local strains are visually represented using the color bar on the right. (**B**) Maximum strains at the 95th percentile calculated from local strain maps (ε_yy_) at 15% bulk strain (*n =* 4–5) (**C**). Correlation analysis between 95th percentile maximum local strains and matched fibril diameter from constructs analyzed using SEM. (* = *p* < 0.05, ** = *p* < 0.01).

**Figure 7 bioengineering-11-01308-f007:**
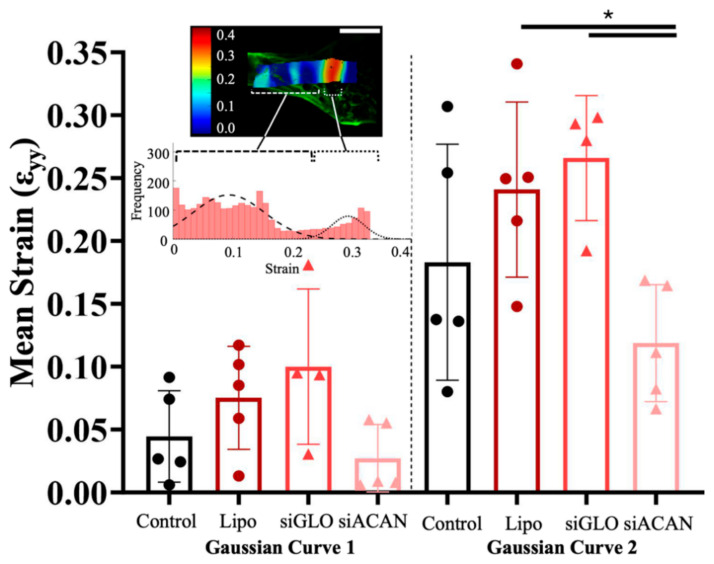
Means of Gaussian curves fitted to strain histograms from untransfected control, Lipofectamine control, siGLO Red control, and siACAN constructs at 15% applied strain. Inset shows a representative confocal elastography image overlayed with a strain map. Underneath, a histogram of strains fitted to two Gaussian curves shows that the first Gaussian curve describes the strains in the bulk tissue and the second describes the higher strain concentrations found at the interface of the bone plug and soft tissue region. Log transformations were taken of these data and analyzed using 1-way ANOVA with Tukey’s multiple comparisons test (*n* = 4–5) (* = *p* <0.05).

## Data Availability

Data is contained within the article or Appendix A.

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
