# Peer review of "siRNA Treatment Enhances Collagen Fiber Formation in Tissue-Engineered Meniscus via Transient Inhibition of Aggrecan Production"

_bioengineering, 2024, doi:10.3390/bioengineering11121308_

Round 1
Reviewer 1 Report
Comments and Suggestions for Authors
(1) Figure 1 (A), fluorescence (right) is missing the label of the scale bar. Please use white font color.
(2) Figure 1 (B) and (C), please explain the reason(s) for the large standard deviations in the discussion.
(3) In Figure 2 (E) caption, please indicate the days of culture for the fiber diameter analysis (e.g., 30 days).
(4) In Figure 4 (B) caption, please indicate the days of culture for the fiber diameter analysis (e.g., 30 days).
(5) In Figure 4, especially (B) and (C), please organize the plots for a better presentation. What is the +/- 0.21 to native? It seems that they should be related to (C), but the labels are overlapping with the axis label of (B). Also, what is the reason to show a break in y-axis in (C)?
(6) It is recommended to include specific mechanical testing procedures in section 2.9. The main question is for Figure 5, where data seem to be reasonable but the scattering of the data points within the groups and the insignificance between the groups may not be supportive to the finding. Please elaborate on this in the discussion in the manuscript.
(7) For readers who are not in authors’ specific field, please can the authors make clear in the manuscript on the tissue constructs? Do they consist of all cultured siRNA cells or are they in layers of normal/siRNA cells?
(8) In Figure 6 (A), the authors mapped the local strain. Since local microstructure will affect the mechanical properties of the tissues, how to the authors view on these tissue constructs in engineering fashion to better tune or improve the mechanical behaviors of siRNA treated collagen fibers in tissue engineering applications?
Reviewer 2 Report
Comments and Suggestions for Authors
The manuscript is interesting but still needs strong revision and correct some data interpretations.
Major remarks:
1. The title of Method section 2.2. needs revision. The methods of sGAG and collagen measurements also should be clearly written not only directed to the citation.
2. Lines 200-204 are copy paste of the same sentences. It is better once clearly to describe methods instead 10 time giving citations.
3. 2.5 section is not informative and can be connected with other analysis methods or diluted at all.
4. 2.9 and 2.10 sections titles – the object of investigation is not clear.
5. Fig. 1. What about the intracellular levels of GAGs and collagens? The authors were measuring only secreted forms, while intracellular levels can be increased in siACAN transfected cells compared to the siGLO or lipofectamine. The extracellular markers do not show the effects of transfection.
6. Fig. 2. The red colors are very similar and difficult to follow what is what.
Fig. 2 legend needs better explanation where is 2D monolayer, and where is 3D construct.
7. Fig. 2 GAGs and collagen was measured inside the cells since it was accumulated.
In Fig. 1 the same parameters were measured as secreted. It cannot be compared in parallel. The Fig.2 B – the explanation of columns is missing.
8. Result sections cannot be without results (as 3.3). Such sections should be connected with other or deleted.
9. Fig. 3 legend needs explanation what color identifies what. If soft gel is in yellow and green, where are collagen fibers?
10. The explanation of Fig. 4 data in the text does not correspond the shown data. What means “Native”? If it is a natural meniscus piece, then the siACAN is missing, while alignment change just a little bit.
Fig. 4 B and C – the explanation of bars is missing.
11. Fig. 5. The effects of siACAN should be compared with Red siRNA controls (siGlo). Otherwise, there is no siACAN effect, and siRNR can be whatever. If so, the siACAN have no effect in Fig.5 and some other figures. To compare siACAN to control/not transfected cells is not correct data interpretation since transfection by itself affects cells. Similar remarks can be addressed to other Figures and data interpretations.
12. The discussion and conclusions also should be corrected according to the revised data interpretations.
Reviewer 3 Report
Comments and Suggestions for Authors
Review of the manuscript entitled: siRNA Treatment Enhances Collagen Fiber Formation in Tissue Engineered Meniscus via Transient Inhibition of Aggrecan Production. The problem of extracellular matrix (ECM) degradation is extremely important, especially in the case of aging. ECM degradation leads to deterioration of skin structure and internal tissues. Moreover, recent findings suggest that ECM degradation may be the cause of neurodegenerative diseases. For this reason, the problem studied by the Authors is of medical essence. Overall, I think that the work is very aesthetic and well prepared, but some corrections should be made before publication.
1. The abstract and introduction are prepared correctly. Generally, it should be noted that the style of the references is incorrect and should be corrected.
2. In my opinion the methodology is very well described! I have suggestions, catalog numbers of key reagents e.g. siRNA or lipofectamine should be added. If siRNA was custom made (no catalog number) please add sequences.
3. Transfections by siRNA are performed in medium without antibiotics and serum. If so, please add to the methodology (unless otherwise).
4. The studies were also performed on animals. Add the bioethics committee approval number (Figures 1,2,3,4).
5. The discussion is correct but there are very long sentences which makes the text difficult to understand. E.g. lines 453-458 – this is one sentence. There are many similar very long sentences in the manuscript. If you want other scientists to understand You, make it easy for them and don't write sentences that are too long. Re-read the discussion and divide the sentences using periods. I understand that you want to show Your language skills, but most scientists in the world are not native English speakers.
6. In my opinion the conclusions are correct, but if possible add a sentence what medical significance your discovery has.
Reviewer 4 Report
Comments and Suggestions for Authors
The manuscript by Lopez et al., entitled “ siRNA Treatment Enhances Collagen Fiber Formation in Tissue Engineered Meniscus via Transient Inhibition of Aggrecan Production ” presented data demonstrating the impact of the transient Inhibition of Aggrecan production on the formation of collagen fibrils The study seem to be interesting and may provide a new information for the reader of the journal. The manuscript is well written. The methods are well described, and the results are clear and support the context of the manuscript. Accordingly, this manuscript can be accepted for the publication in the present form
Round 2
Reviewer 1 Report
Comments and Suggestions for Authors
Thank you for addressing my comments. They were all reasonable and professional within the context of the review. Best wish to the progress of the research.
Reviewer 2 Report
Comments and Suggestions for Authors
The manuscript has been improved; however, some minor remarks remain to further clarify it.
1. 2,3 part „Tissue Engineered Construct Preparation” would strongly benefit from the scheme, showing the type of used constructs. Now it is difficult to follow and to understand those types of constructs.
2. Fig. 3 needs an arrows showing the structures the authors are talking about.
3. The 3,2 paragraph is not corrected as the authors state in their answers to the reviewer’s comments.
The authors are saying that „This paragraph was revised as follows:”, however it is not. The corrections made are not present in the article's text.
“Although no statistically significant changes in the bulk mechanical properties were observed between the three control groups and the siACAN group due to the variability of the data, the averages of the Lipofectamine, siGLO, and siACAN groups showed a slight upward trend compared to control. Notably, siACAN constructs demonstrated a 63% higher ultimate load than control (p=0.45). In addition, the average toughness, Young’s modulus, and UTS of these three groups also displayed a slight increasing trend compared to untransfected control. Toughness, Young’s modulus, and UTS were increased in siACAN constructs by 64% (p=0.3), 125% (p=0.37), and 117% (p=0.46), respectively compared to untransfected control. Within the siACAN group, the bulk mechanical data suggest the presence of two distinct subpopulations: one resembling the control group and another exhibiting enhanced mechanical properties. While the overlap of these subpopulations results in overall mechanical properties that are not significantly different to controls, the presence of a subset with increased mechanical properties is noteworthy.”
4. All corrections made in the manuscript should be clearly marked and align with the authors' responses to the reviewers' comments; otherwise, it indicates that the article was not properly corrected.
